# Clinical Efficacy and Safety of an Automatic Closed-Suction System in Mechanically Ventilated Patients with Pneumonia: A Multicenter, Prospective, Randomized, Non-Inferiority, Investigator-Initiated Trial

**DOI:** 10.3390/diagnostics14111068

**Published:** 2024-05-21

**Authors:** Dong-Hyun Joo, Hyo Chan Park, Joon Han Kim, Seo Hee Yang, Tae Hun Kim, Hyung-Jun Kim, Myung Jin Song, Sung Yoon Lim, Sung A Kim, Hee Won Bae, Yoon Hae Ahn, Si Mong Yoon, Jimyung Park, Hong Yeul Lee, Jinwoo Lee, Sang-Min Lee, Jung Chan Lee, Young-Jae Cho

**Affiliations:** 1Division of Pulmonary and Critical Care Medicine, Department of Internal Medicine, Seoul National University College of Medicine, Seoul National University Bundang Hospital, Seongnam 13620, Republic of Korea; jdhyun2000@snu.ac.kr (D.-H.J.); hyochanpark@snu.ac.kr (H.C.P.); chemokim@gmail.com (J.H.K.); 0227ya@naver.com (S.H.Y.); kimth1023@naver.com (T.H.K.); dr.hjkim@snu.ac.kr (H.-J.K.); mjsong8705@gmail.com (M.J.S.); nucleon727@gmail.com (S.Y.L.); 2Department of Medical Device Development, Seoul National University College of Medicine, Seoul 03080, Republic of Korea; 3Division of Pulmonary and Critical Care Medicine, Department of Internal Medicine, Seoul National University College of Medicine, Seoul National University Hospital, Seoul 03080, Republic of Korea; 49031@snuh.org (S.A.K.); baehw1064@snu.ac.kr (H.W.B.); boxer0707@hanmail.net (J.P.); realrain7@gmail.com (J.L.); sangmin2@snu.ac.kr (S.-M.L.); 4Department of Critical Care Medicine, Seoul National University Hospital, Seoul 03080, Republic of Korea; yahn14@gmail.com (Y.H.A.); paramount_05@naver.com (S.M.Y.); takumama@naver.com (H.Y.L.); 5Department of Biomedical Engineering, Institute of Medical and Biological Engineering, Medical Research Center, Seoul National University College of Medicine, Seoul 03080, Republic of Korea; ljch@snu.ac.kr

**Keywords:** endotracheal suctioning, critical care, mucosal secretions, intensive care unit

## Abstract

Endotracheal suctioning is an essential but labor-intensive procedure, with the risk of serious complications. A brand new automatic closed-suction device was developed to alleviate the workload of healthcare providers and minimize those complications. We evaluated the clinical efficacy and safety of the automatic suction system in mechanically ventilated patients with pneumonia. In this multicenter, randomized, non-inferiority, investigator-initiated trial, mechanically ventilated patients with pneumonia were randomized to the automatic device (intervention) or conventional manual suctioning (control). The primary efficacy outcome was the change in the modified clinical pulmonary infection score (CPIS) in 3 days. Secondary outcomes were the frequency of additional suctioning and the amount of secretion. Safety outcomes included adverse events or complications. A total of 54 participants, less than the pre-determined number of 102, were enrolled. There was no significant difference in the change in the CPIS over 72 h (−0.13 ± 1.58 in the intervention group, −0.58 ± 1.18 in the control group, *p* = 0.866), but the non-inferiority margin was not satisfied. There were no significant differences in the secondary outcomes and safety outcomes, with a tendency for more patients with improved tracheal mucosal injury in the intervention group. The novel automatic closed-suction system showed comparable efficacy and safety compared with conventional manual suctioning in mechanically ventilated patients with pneumonia.

## 1. Introduction

Endotracheal suctioning constitutes a frequent, essential procedure in mechanically ventilated patients [1,2]. Closed endotracheal suctioning is the most effective method for clearing excessive mucosal secretions [3] and is typically performed at 1 to 2 h intervals in intensive care units (ICUs) [4]. The procedure is labor intensive and the associated risks include serious complications, such as bleeding, infection, hypoxia, and cardiovascular issues, if incorrectly performed without following clinical practice guidelines [2,3,4,5]. Due to the aging population, the number of patients with acute and chronic respiratory failure has rapidly increased and, due to shortages of nurses, there is increasing demand for alternatives to manual suctioning, especially after the coronavirus disease (COVID-19) pandemic [6,7].

To address these issues, a novel A-1000 automated closed-suction device (L-MECA Co., Seoul, Republic of Korea) was developed [8] for suction regulation, with a consistent suction pressure, duration, and intubation depth throughout the procedure. The A-1000 device utilizes a long suction catheter and integrated antibacterial disinfection that effectively prevents the entry of infectious agents into the device. This device can potentially alleviate the work burden on healthcare providers, minimize endotracheal suction-related complications, and mitigate the risks of cross-contamination between patients and healthcare providers. In 2019, a pilot study based on tracheal mucosal injury reported comparable safety of the automatic closed-suction system with conventional manual closed suction, in five mechanically ventilated ICU patients [8]. The A-1000 device is expected to be most useful and necessary in ICU settings, with a high endotracheal suction burden. However, there are limited clinical data on the therapeutic application of the A-1000 device in mechanically ventilated patients with acute or chronic respiratory failure.

We posited that compared with the conventional closed-suction system, the automated closed-suction system would yield non-inferior efficacy and safety outcomes in mechanically ventilated patients with pneumonia. Thus, this study aimed to evaluate the clinical efficacy and safety of a novel automatic closed-suction system in mechanically ventilated patients with pneumonia for ≥72 h.

## 2. Materials and Methods

### 2.1. Study Design and Population

This multicenter, prospective, randomized, non-inferiority investigator-initiated trial was conducted in the medical ICU of tertiary referral hospitals from May 2021 to December 2022. Adult patients (age ≥ 19 years) diagnosed with pneumonia and requiring mechanical ventilation for >72 h were enrolled. We excluded patients who could not undergo bronchoscopy; were expected to be weaned-off mechanical ventilation in 72 h or had an Acute Physiology and Chronic Health Evaluation II (APACHE II) score > 30 and were expected to expire within 72 h; needed mechanical ventilation with FiO_2_ > 80% or P_plat_ > 30 cmH_2_O to maintain adequate oxygenation; or had significant immunodeficiency. More detail on the criteria is provided in Appendix A. Written informed consent was obtained from the participants or their legally authorized representatives. This study was approved by the Institutional Review Board (IRB) at each participating hospital (IRB No. 2102-666-002 in SNUBH and No. 2104-040-1210 in SNUH). The trial was pre-registered on cris.nih.go.kr (No. KCT0006608).

### 2.2. Intervention and Randomization

All eligible participants were randomly allocated to the automatic closed-suction (intervention) group or the conventional manual closed-suction (control, routine clinical manual suction) group for 72 ± 3 h. An independent clinical research organization undertook 1:1 participant allocation using a sealed-envelope method with permuted block randomization (stratified by the modified Clinical Pulmonary Infection Score (CPIS) of 7 and the research center), designed by an independent statistician. Manual suction was performed every 120 min with 300 mmHg of suction pressure, although this was not necessarily a fixed cycle or pressure as it depended on the patient’s condition and clinical situation, and additional manual suction was allowed based on clinical necessity. In the automated closed-suction group, the A-1000 device (see Appendix A) [8], an automated closed-suction system, licensed and commercially available in South Korea, was used. The nurse in charge determined the catheter’s inner diameter and the depth of insertion, based on the patient’s condition. Automatic suction was performed with just one press of a button, and the frequency was equal to that of the control group, with additional automatic suction permitted, as in the control group. At baseline and study completion, the CPIS and grade of tracheal mucosal injury as assessed using bronchoscopy was calculated for all participants. Throughout the study, the suction frequency, secretion amount, and complications or adverse events were recorded.

### 2.3. Outcomes

The primary efficacy outcome was the change in the CPIS after 72 ± 3 h. The secondary efficacy outcomes were the CPIS at both the baseline and after 72 ± 3 h; the number of suctions performed, in addition to the protocol-specified suction for 72 ± 3 h; the total amount of secretion (cc) collected for 72 ± 3 h; and the device satisfaction survey after 72 ± 3 h on a 10-point numerical rating scale (only in the intervention group). The safety outcomes included treatment-emergent adverse events (TEAEs); the complication rate (%), including hypoxemia, atelectasis, infection, hemoptysis, and tracheal mucosal injury; the grade of tracheal mucosal injury, which was evaluated through a bronchoscopy image just below the endotracheal tube at day 0 and day 3 by independent bronchoscopists, using a previously developed 5-point scale (0: normal; 1: erythema or edema; 2: erosion; 3: hemorrhage; and 4: ulceration or necrosis) [9]; the device malfunction rate (%), evaluated by designated nurses (only in the intervention group); and bacterial contamination at the catheter connection site.

### 2.4. Sample Size Calculation

As data for the automatic closed-suction system in mechanically ventilated patients with pneumonia were lacking, we considered the change in the CPIS as a surrogate efficacy marker of the suction systems and indirectly used the change in the CPIS in the conventional manual closed-suction system to estimate the sample size, using a research hypothesis based on the mean change and standard deviation from two different studies [10,11]. We estimated that the enrollment of 41 participants in each group would provide 80% power and a significance level of 2.5% using a non-inferiority margin of 0.84. We planned to enroll 102 participants, considering a 20% dropout rate (described in detail in Appendix A).

### 2.5. Statistical Analysis

The efficacy outcomes were analyzed in the full analysis (FA) and per protocol (PP) sets. The FA set included all participants who underwent randomization and evaluation of the efficacy outcome at least once. The PP set included all participants who completed the trial without significant protocol violations. The final analysis of the efficacy outcome was conducted in the PP set. The safety outcomes were analyzed in a safety set, which included all participants who underwent randomization. For all the missing values, complete case analysis was conducted without statistical imputation. The continuous variables are presented as the median + interquartile range or the mean ± standard deviation. The categorical variables are presented as the frequency (proportion). In regard to the primary efficacy outcome, the confidence interval (CI) was calculated for the intergroup difference in the change in the CPIS after 72 ± 3 h. If the upper limit of the calculated 97.5% one-sided CI was less than the non-inferiority margin of 0.84, the intervention group was judged to be non-inferior compared with the control group. For the other outcomes, an independent two-sample *t*-test or Wilcoxon rank sum test was performed for the continuous variables, and the chi-square or Fisher’s exact test was conducted for the categorical variables. All the analyses, except the primary efficacy outcome, were two sided; *p* values < 0.05 indicated statistical significance. All the analyses were conducted using SAS, version 9.4 (SAS institute, Cary, NC, USA).

## 3. Results

### 3.1. Participant Characteristics

From May 2021 to December 2022, 57 participants were enrolled at SNUBH and SNUH. The trial was prematurely terminated due to recruitment difficulties during the COVID-19 pandemic. All of the 57 screened participants who met the inclusion and exclusion criteria for the trial were randomly assigned to the intervention (*n* = 28) and control (*n* = 29) groups. After excluding three patients who did not receive the scheduled treatment (two and one patient in the intervention and control groups, respectively), 54 participants (26 and 28 patients in the intervention and control groups, respectively) were allocated to the safety set. All 54 participants in the safety set underwent an evaluation of the efficacy outcomes at least once and comprised the FA set; however, seven participants (one patient with atrial fibrillation and two with cardiac arrest in the intervention group, four who were extubated within 3 days in the control group) dropped out, leaving 47 participants in the PP set (23 and 24 patients in the experimental and control groups, respectively; Figure 1). At the baseline, the FA set had evenly distributed demographics, including age and sex, in the study groups. The median duration of pneumonia at enrollment was 4.0 (1.0–9.0) days, and the APACHE II score at the baseline was 20.33 ± 4.84; 46 of 54 (85.19%) patients had diffuse lung infiltration on their chest radiography and 30 of 54 (55.56%) were diagnosed with acute respiratory distress syndrome (ARDS) at the baseline (Table 1). No significant intergroup difference in the other variables, including vital signs, laboratory findings, and medical history, were present (Appendix A).

### 3.2. Efficacy Outcomes

In the PP set, there was no significant intergroup difference in the changes in the CPIS after 72 ± 3 h (−0.13 ± 1.58 and −0.58 ± 1.18 in the intervention and control groups, respectively, *p* = 0.866; Figure 2A), with an upper limit in the calculated 97.5% one-sided CI of 1.27, which exceeded the pre-specified non-inferiority margin of 0.84 and indicated that the intervention group did not demonstrate non-inferiority compared with the control group (Table 2). The FA and PP sets had the same participants; therefore, the analysis of the primary efficacy outcomes of the FA and PP sets were identical (see Appendix A). The CPIS at the baseline and after 72 ± 3 h in the PP set and the intergroup difference in the CPIS at each time point did not differ significantly (3.39 ± 1.78 vs. 3.75 ± 1.92 at baseline, 3.26 ± 1.48 vs. 3.17 ± 1.5 at 72 ± 3 h). The rate of improvement in the CPIS, with improvement and no improvement, was defined as a ≥1 point increase or <1 point increase, respectively, which after 72 ± 3 h was 39.13% and 54.17% for the intervention and control groups, respectively, without a significant intergroup difference (*p* = 0.385) (Table 2).

In the PP set, the number of suctions performed in addition to those specified in the protocol for 72 ± 3 h was 8.2 6 ± 6.72 and 6.38 ± 6.35 times in the intervention and control group, respectively, which was non-significantly higher in the intervention group (*p* = 0.189). Regarding the total amount of secretion (cc) collected after 72 ± 3 h in the PP set, the intervention group showed a similar secretion volume compared with the control group (86.0 (42.0–247.0) cc vs. 147.5 (48.5–207.5) cc, *p* = 0.835; Figure 2B). A similar tendency for additional suction frequency and total secretion collected in the analysis of the FA set was noted. In the device satisfaction survey undertaken by the nurse in charge after 72 ± 3 h, the mean score was 5.88 ± 1.05 out of 10 (Table 2).

### 3.3. Safety Outcomes

During the study, three TEAEs were observed, all within the intervention group: one atrial fibrillation (mild) and two cardiac arrests (severe). These TEAEs were determined to be unrelated to the investigated medical device and were not classified as device-related adverse reactions (Table 3). No participant had complications, such as hypoxemia, atelectasis, infection, hemoptysis, and tracheal mucosal injury. No significant intergroup difference was noted in the grade of the tracheal mucosal injury, evaluated bronchoscopically at days 0 and 3 (see Appendix A), or in the tracheal mucosal injury incidence rate (3.85% and 3.57% in the intervention and control groups, respectively), which was defined as an increase in the grade of tracheal mucosal injury by more than 1 point at day 3 compared with that on day 0. Conversely, when tracheal mucosal injury improvement was defined as a decrease in the grade of tracheal mucosal injury by 1 point or more at day 3 compared with that at day 0, more participants in the intervention group experienced an improvement in their mucosal injury compared with those in the control group (52.17% vs. 25.00%); although the intergroup difference was not significant (Table 3, Appendix A). The device malfunction rate was 3.85% (1 of 26) in the intervention group, with one occurrence of malfunction due to an error in measuring the secretion amount (Table 3). To determine the presence of bacterial contamination at the catheter connection site after using the intervention device, bacterial culture tests were performed. Among the 23 participants in the intervention group, bacterial contamination with *Pseudomonas putida* and *Enterococcus faecium* was confirmed in two individuals (8.70%).

## 4. Discussion

To our knowledge, this is the first multicenter, prospective, randomized, non-inferiority investigator-initiated trial that evaluated the efficacy and safety of an automatic closed-suction system in mechanically ventilated patients with pneumonia. In this study, we reported comparable efficacy, measured by the CPIS, although the results did not statistically satisfy the non-inferiority and satisfactory safety outcomes assessed based on adverse events, complications, and tracheal mucosal injury, when the automated system was applied for 72 ± 3 h in mechanically ventilated patients with pneumonia compared with the conventional manual closed-suction system.

Previous studies on endotracheal suctioning with a closed-suction system compared with a conventional “open” suction system have mainly focused on the clinical outcomes [1]. Despite the relatively less effective secretion removal [12,13,14], the closed-suction system showed better outcomes with regard to physiologic parameters, including heart rate, blood pressure, oxygen saturation, arrhythmia, and intracranial pressure, in many studies [12,13,15]. Given the physiologic merit of minimizing the de-recruitment of lung volume provoked by disconnection, the closed-suction system was adopted as the recommended method of suctioning, especially in adult patients with high F_i_O_2_ or PEEP and in ARDS patients [3]. However, no other researcher, until recently, had investigated the therapeutic application of automatic closed-suction systems in randomized controlled trials.

The current study presents diverse, expanded clinical outcomes of automatic closed-suction systems compared with those reported in a previous pilot study [8], wherein the system was applied for only 24 h in a limited sample of five participants. In the present study, the application of the system was extended to 72 ± 3 h and included a larger sample of 47 participants in the PP set, including 23 patients in the experimental group. Although the previous study reported the performance of the medical device in a dummy model only, the current study evaluated the efficacy of the device via various clinical outcomes, including the change in the CPIS, the total collected secretion, and the suctioning frequency. Lastly, the previous study presented limited safety outcomes indicated by the tracheal mucosal injury, whereas in the current study, various clinically important safety outcomes were demonstrated, including adverse events, complications, the device malfunction rate, and the presence of bacterial contamination at the catheter connection site. Furthermore, the medical device was applied without safety issues to patients who required prone positioning with moderate-to-severe ARDS. The characteristics of the current trial enable the addressal of the medical device’s applicability in the clinical setting, based on its comparable efficacy and potential for alleviating the workload of healthcare providers without increasing complications.

In the analysis of the primary efficacy outcome, although there was no significant intergroup difference in the change in the CPIS after 72 ± 3 h (*p* = 0.866), the non-inferiority of the intervention group compared to that of the control group was not proved. One possible explanation involves the difference between the predicted and actual severity of pneumonia. The baseline CPIS of the participants in this clinical trial was 3.69 ± 1.84, which was lower than the baseline CPIS reported in studies used as a reference for sample size estimation, Cho et al. [10] and Lee et al. [11] reported a mean score of 7.80 ± 1.2 and 6.89 ± 1.35, respectively. It may have been challenging to observe a significant change in the CPIS after 3 days in this particular trial. Furthermore, during the COVID-19 pandemic, healthcare professionals from the medical ICUs were reassigned to COVID-19 wards, and some beds in the ICUs were converted for COVID-19 care. Consequently, the reduced number of patients in the medical ICUs led to slower participant enrollment; the enrolment of only 47 patients directly resulted in the underpowered status to verify the primary efficacy outcomes.

Concerns exist about the increased risk of ventilator-associated pneumonia (VAP) or bacterial colonization with a closed-suction system compared with an open-suction system [4]. However, most previous studies reported no significant difference in the incidence of VAP or bacterial colonization [16,17,18,19,20]. Herein, we reported two participants (2/23, 8.7%) with bacterial colonization of the catheter connection site with *P. putida* and *E. faecium* in the intervention group. Before and after the study, these bacteria were not identified in the respiratory sputum of the participants who used the intervention device from which each bacteria emerged. As bacteria from previous patients may remain in the intervention device without being disinfected, the result of the respiratory sputum culture of the patients who used the intervention device, wherein each bacterium was detected before and after the identification, was analyzed. However, those same organisms were not identified in those patients.

Despite the aforementioned novel findings in the present study, there are some limitations that should be acknowledged. First, due to the COVID-19 pandemic, the trial was prematurely terminated and failed to completely enroll participants, and this led to the recruitment of a study population that was underpowered to verify non-inferiority. However, as the change in the CPIS may not directly reflect an improvement in pneumonia [21] and the change could be influenced by other variables besides endotracheal aspiration, the CPIS may not constitute an accurate surrogate to evaluate the efficacy of the endotracheal suction method. Second, although we extended the use of the medical device compared with that in the pilot study, 3 days may be insufficient to evaluate efficacy or safety outcomes. However, the intervention device was not inferior to the manual suction method, at least in terms of the safety outcomes. The innate methodological advantage of the automatic suction technique, in regard to decreasing the workload and the risk of cross-contamination, ensures that the medical device is sufficiently useful for clinical use in mechanically ventilated patients with pneumonia, who constitute a high burden in term of endotracheal suctioning. Lastly, in association with the second limitation, we could not provide more clinically important outcomes, including the VAP occurrence, mortality, duration of mechanical ventilation, and ICU length of stay, because of the relatively short duration of the study. Therefore, further studies are needed with larger populations and longer-duration interventions to determine the clinically important outcomes.

## 5. Conclusions

The novel automatic closed-suction system failed to show a non-inferior primary efficacy outcome in the change in the CPIS after 3 days. However, the device showed comparable efficacy in regard to the total amount of collected secretion, the additional frequency of suctioning, and equivalent safety, including adverse events, complications, and the tracheal mucosal injury grade, compared with the conventional manual suction system, in patients with pneumonia who required mechanical ventilation. The system could serve as a valuable alternative approach, given the promising results in this study in mechanically ventilated ICU patients. Further large-scale studies are needed to confirm the clinical efficacy and safety of the system in ICU patients with a high burden in regard to endotracheal suctioning.

## Figures and Tables

**Figure 1 diagnostics-14-01068-f001:**
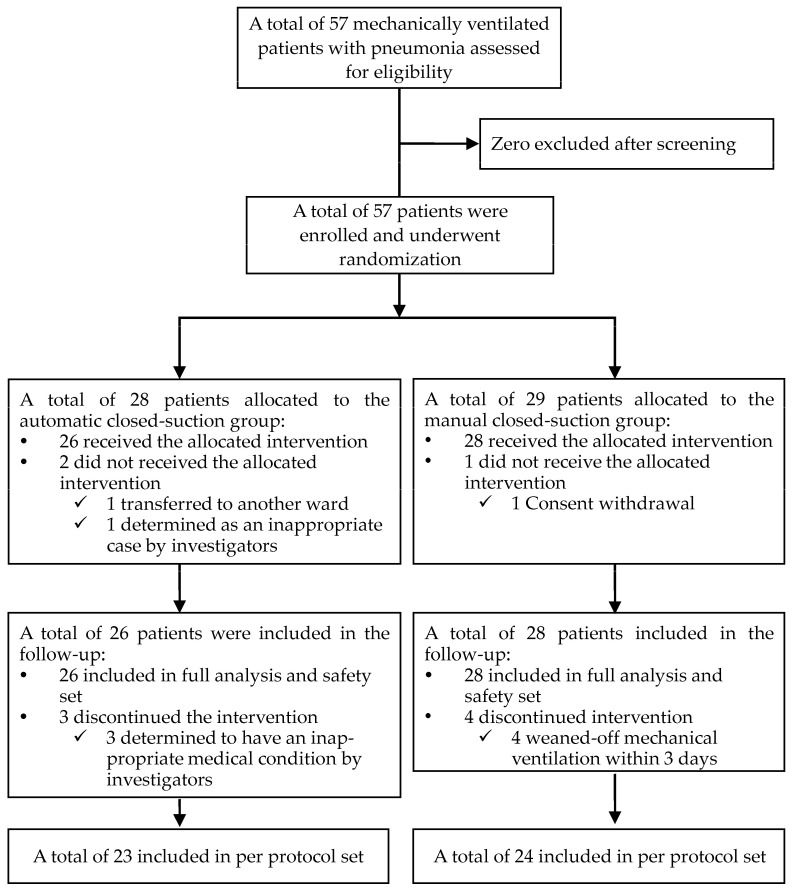
Flowchart depicting the selection of the study population.

**Figure 2 diagnostics-14-01068-f002:**
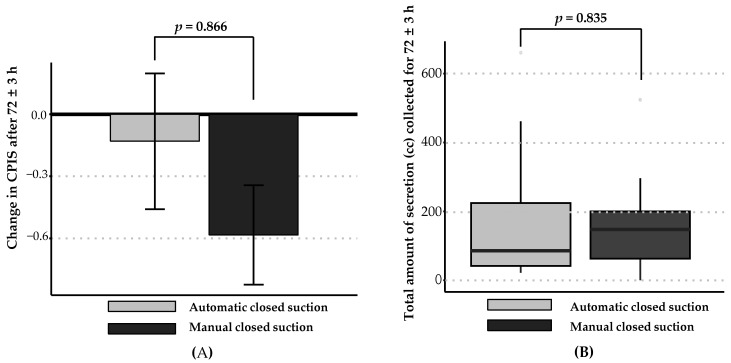
Efficacy outcomes of the per protocol (PP) set analysis. (**A**) The change in CPIS after 72 ± 3 h is shown for each group. The change in CPIS after 72 ± 3 h was −0.13 ± 1.58 in the automatic closed-suction group (gray) and −0.58 ± 1.18 in the manual closed-suction group (black). (**B**) The total amount of secretions (cc) collected after 72 ± 3 h is shown for each group. The automatic closed-suction group had comparable amounts of collected secretion during the study period as the manual closed-suction group (162.00 ± 175.76 cc vs. 142.75 ± 121.44 cc, *p* = 0.835).

**Table 1 diagnostics-14-01068-t001:** Participant characteristics at baseline for the full analysis set.

Variables	Automatic Closed Suction	Manual Closed Suction	Total	
	(*n* = 26)	(*n* = 28)	(*n* = 54)	
Age, years	66.1 ± 18.1	71.5 ± 10.9	68.9 ± 14.9	
Sex, male	22 (84.6%)	22 (78.6%)	44 (81.5%)	
Height, cm	164.0 ± 7.6	165.2 ± 9.4	164.7 ± 8.5	
Weight, kg	61.4 ± 12.6	59.4 ± 15.6	60.4 ± 14.2	
Duration of pneumonia, days	4.0 (1.0–9.0)	5.0 (2.5–11.0)	4.0 (1.0–9.0)	
Duration in ICU, days	2.0 (2.0–5.0)	2.0 (1.0–6.0)	2.0 (1.0–6.0)	
APACHE II score	19.7 ± 5.6	20.9 ± 4.1	20.3 ± 4.8	
Chest radiography				
Diffuse infiltration	22 (84.6%)	24 (85.7%)	46 (85.2%)	
Localized infiltration	4 (15.4%)	4 (14.3%)	8 (14.8%)	
ARDS	15 (57.7%)	15 (53.6%)	30 (55.6%)	
Hemodynamic variables				
Mean blood pressure, mmHg	87 ± 14	90 ± 16	89 ± 15	
Heart rate, /min	92 ± 22	97 ± 22	95 ± 22	
Respiratory rate, /min	21 ± 5	22 ± 4	22 ± 5	
Body temperature, °C	36.9 ± 0.6	36.8 ± 0.6	36.9 ± 0.6	
SpO_2_, %	97 ± 2	97 ± 3	97 ± 3	

APACPE II: Acute Physiology and Chronic Health Evaluation II, ARDS: acute respiratory distress syndrome. Continuous variables are presented as the mean ± standard deviation or the median (interquartile range), and the categorical variables are presented as frequency (proportion).

**Table 2 diagnostics-14-01068-t002:** Efficacy outcomes for the per protocol set of participants.

Variables	Automatic Closed Suction	Manual Closed Suction	*p*-Value
	(*n* = 23)	(*n* = 24)	
Primary efficacy outcome			
Change in CPIS = (CPIS after 72 h − baseline CPIS)			0.866
Number	23	24	
Mean ± SD	−0.13 ± 1.58	−0.58 ± 1.18	
Mean difference (intervention—control) (97.5% confidence interval for difference)	0.45 (−∞, 1.27)		
Secondary efficacy outcomes			
Modified CPIS at baseline	3.39 ± 1.78	3.75 ± 1.92	0.531
Modified CPIS at 72 ± 3 h	3.26 ± 1.48	3.17 ± 1.58	0.760
* Improvement rate in modified CPIS	9 (39.13%)	13 (54.17%)	0.385
Number of suctions performed in addition to those specified in the protocol	7.0 (3.5–12.0)	4.0 (2.0–7.5)	0.189
Total amount of secretions collected, cc	86.0 (42.0–247.0)	147.5 (48.5–207.5)	0.835
^†^ Device satisfaction survey after 72 h	5.88 ± 1.05		

CPIS: Modified Clinical Pulmonary Infection Score, SD: standard deviation. Continuous variables are presented as the mean ± standard deviation or the median (interquartile range), and categorical variables are presented as frequency (proportion). * Classified as “improvement” if the modified CPIS improved by 1 point or more after 72 ± 3 h. ^†^ Device satisfaction survey after 72 ± 3 h was evaluated using a 10-point numerical rating scale (the higher the score, the higher the satisfaction).

**Table 3 diagnostics-14-01068-t003:** Safety outcomes in the safety set analysis.

Variables	Automatic Closed Suction	Manual Closed Suction	Total
	(*n* = 26)	(*n* = 28)	(*n* = 54)
Treatment-emergent adverse events	3 (11.54%)	0 (0.00%)	3 (5.56%)
Mild (atrial fibrillation)	1 (3.85%)	0 (0.00%)	1 (1.85%)
Moderate	0 (0.00%)	0 (0.00%)	0 (0.00%)
Severe (cardiac arrest)	2 (7.69%)	0 (0.00%)	2 (3.70%)
Device-related adverse reactions	0 (0.00%)	0 (0.00%)	0 (0.00%)
Complication rate (%)	0 (0.00%)	0 (0.00%)	0 (0.00%)
Tracheal mucosal injury incidence (%)	1 (3.85%)	1 (3.57%)	2 (3.70%)
	*p* = 1.000		
Tracheal mucosal injury improvement (%)	12 (46.15%)	6 (21.43%)	18 (33.33%)
	*p* = 0.102		
Device malfunction rate	1 (3.85%)		

Incidence of tracheal mucosal injury was defined as an increase in the grade of tracheal mucosal injury by more than 1 point at day 3 compared with that at day 0. Improvement of tracheal mucosal injury was defined as a decrease in the grade of tracheal mucosal injury by 1 point or more at day 3 compared with that at day 0. Categorical variables are presented as frequency (proportion).

## Data Availability

The de-identified data used and analyzed during the current study are available from the corresponding author upon reasonable request. The request should be directed to the corresponding author via email: lungdrcho@snu.ac.kr.

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
