# Peer review of "Clinical Efficacy and Safety of an Automatic Closed-Suction System in Mechanically Ventilated Patients with Pneumonia: A Multicenter, Prospective, Randomized, Non-Inferiority, Investigator-Initiated Trial"

_diagnostics, 2024, doi:10.3390/diagnostics14111068_

Round 1

Reviewer 1 Report

Comments and Suggestions for Authors

Dear authors,

thank you for the opportunity to read and review the manuscript.

The topic is interesting and the paper is well written.

General comments

Closed suction system is the recommended suction methods in intubated patients in ICU; strict evidence is lacking on VAP incidence related to automated or manual closed suction system.

Specific comments

Line 55. Which antibacterial is used for disinfection? How is manual closed suction system cleaned after suction?

Were the suction systems of interventional group used immediately after intubation in the study population?   

Results

At enrollment and during the study period there are data on blood gas analysis, mostly PaO2 and PaCO2, that can be useful to evaluate differences in the risk of dereclutation, hypoxia, hypoventilation?

Line 252 - Why bacterial contamination evaluation was performed only on intervention group? 

Line 295- How was the reduction in labor of healthcare providers  assessed? 

Reviewer 2 Report

Comments and Suggestions for Authors

There are concerns regarding rigor and inappropriate conclusions. There is no enough data to support that the automatic device present “no inferiority”.

There is inconsistency between the text, figures, and tables. For instance, which is the real number of total amounts of secretion? Text line 206 has (162.00±175.76 cc vs. 142.75±121.44 cc, p=0.835, Figure. 2B), values that are not reflected in the figure, in which higher “Y” axis value is two hundred. Additionally, table numbers for that metric are 86.0 (42.0 – 247.0) 147.5 (48.5 – 207.5). It is impossible to believe in the rigor of this study.
